# Deeper Insights into Weight Sharing in Neural Architecture Search

## Abstract

With the success of deep neural networks, Neural Architecture Search (NAS) as a way of automatic model design has attracted wide attention. As training every child model from scratch is very time-consuming, recent works leverage weight-sharing to speed up the model evaluation procedure. These approaches greatly reduce computation by maintaining a single copy of weights on the super-net and share the weights among every child model. However, weight-sharing has no theoretical guarantee and its impact has not been well studied before. In this paper, we conduct comprehensive experiments to reveal the impact of weight-sharing: (1) The best-performing models from different runs or even from consecutive epochs within the same run have significant variance; (2) Even with high variance, we can extract valuable information from training the super-net with shared weights; (3) The interference between child models is a main factor that induces high variance; (4) Properly reducing the degree of weight sharing could effectively reduce variance and improve performance.

## 1 Introduction

Learning to design neural architectures automatically has aroused wide interests recently due to its success in many different machine learning tasks. One stream of neural architectures search (NAS) methods is based on reinforcement learning (RL) (Zoph & Le, 2016; Zoph et al., 2018; Tan et al., 2019), where a neural architecture is built from actions and its performance is used as reward. This approach usually demands considerable computation power — each search process takes days with hundreds of GPUs. Population based algorithms (Gaier & Ha, 2019; Liang et al., 2018; Jaderberg et al., 2017) are another popular approach for NAS, new trials could inherit neural architecture from better performing ones as well as their weights, and mutate the architecture to explore better ones. It also has high computation cost.

To speed up the search process, a family of methods attracts increasing attention with greatly reduced computation (Pham et al., 2018; Liu et al., 2018c; Bender et al., 2018). Instead of training every child model, they build a single model, called *super-net*, from neural architecture search space, and maintain a single copy of weights on the super-net. Several training approaches have been proposed on this model, *e.g.,* training with RL controller (Pham et al., 2018), training by applying dropout (Bender et al., 2018) or architecture weights on candidate choices (Liu et al., 2018c). In these approaches, weight-sharing is the key for the speedup. However, weight sharing has no theoretical guarantee and its impact has not been well studied before. The directions of improving such methods would be more clear if some key questions had been answered: 1) How far is the accuracy of found architecture from the best one within search space? 2) Could the best architecture be stably found in multiple runs of search process? 3) How does weight sharing affect the accuracy and stability of the found architecture?

In this paper, we answer the above-mentioned questions using comprehensive experiments and analysis. To understand the behavior of weight sharing approaches, we use a small search space, which makes it possible to have ground truth for comparison. It is a simplified NAS problem, therefore, making it easy to show the ability of the NAS algorithms with weight sharing. As a result, we find that the rank of child models is very unstable in different runs of the search process, and also very different from ground truth. In fact, the instability [1] commonly exists not only in different runs,

---

[1] In this paper, we use instability and variance interchangeably.

but also in consecutive training epochs within the same run. Also worthy of note, in spite of high variance, we can extract statistical information from the variance, the statistics can be innovatively leveraged to prune search space and improve the search result.

To further understand where the variance comes from, we record and analyze more metric data from the experiments. It is witnessed that some child models have interference with each other, and the degree of this interference varies depending on different child models. At the very end of the super-net training, training each child model in one mini-batch can make this model be the best performing one on the validation data. Based on the insights, we further explore partial weight sharing, that is, each child model could selectively share weights with others, rather than all of them sharing the same copy of weights. It can be seen as reduced degree of weight sharing. One method we have explored is sharing weights of common prefix layers among child models. Another method is to cluster child models into groups, each of which shares a copy of weights. Experiment results show that partial weight sharing makes the rank of child models more stable and becomes closer to ground truth. It implies that with proper degree or control of weight sharing, better child models can be more stably found.

To summarize, our main contributions are as follows:

- We define new metrics for evaluating the performance of the NAS methods based on weight sharing, and propose a down-scaled search space which makes it possible to have a deeper analysis by comparing it with ground truth.
- We design various experiments, and deliver some interesting observations and insights. More importantly, we reveal that valuable statistics can be extracted from training the super-net, which can be leveraged to improve performance.
- We take a step further to explain the reasons of high variance. Then we use decreased degree of weight sharing, which shows lower variance and better performance, to support the reasoning.

## 2  RELATED WORKS

Neural Architecture Search (NAS) is invented to relieve human experts from laborious job of engineering neural network components and architectures by automatically searching optimal neural architecture from a human-defined search space. Arguably, the recent growing interesting in NAS research begins from the work by Zoph and Le (Zoph & Le, 2016) where they train a controller using policy gradients (Williams, 1992) to discover and generate network models that achieve state-of-the-art performance. Following these works, there is a growing interest in using RL in NAS (Pham et al., 2018; Baker et al., 2017; Tan et al., 2019; Zoph et al., 2018). There have also been studies in evolutionary approaches (Real et al., 2019; 2017; Miikkulainen et al., 2017; Xie & Yuille, 2017; Liu et al., 2018b). Most of these works still demand high computational cost that is not affordable for large networks or datasets.

**Weight sharing approaches**   Weight sharing means sharing architecture weights among different components or models. Pham et al. (2018) combined this approach with previous work of NAS (Zoph & Le, 2016) and proposed Efficient Neural Architecture Search (ENAS), where a super-net is constructed which contains every possible architecture in the search space as its child model, and thus all the architectures sampled from this super-net share the weights of their common graph nodes. It significantly reduces the computational complexity of NAS by directly training and evaluating sampled child models directly on the shared weight. After the training is done, a subset of child models is chosen and they are either finetuned or trained from scratch to get the final model.

Many follow-up works leverage weight sharing as a useful technique that can be decoupled from RL controllers, including applying dropout on candidate choices (Bender et al., 2018), converting the discrete search space into a differentiable one (Liu et al., 2018c; Wu et al., 2018; Xie et al., 2019), searching via sparse optimization (Zhang et al., 2018), and directly searching for child models for large-scale target tasks and hardwares (Cai et al., 2019).

**Previous studies on stability of weight sharing**   All the weight-sharing approaches are based on the assumption that the rank of child models obtained by evaluating a child model of the trained super-net is valid, or at least, capable of finding one of the best child models in the search space.

However, this assumption does not generally hold. For example, Guo et al. (2019) believed that child models are deeply coupled during optimization, causing high interference among each other. Sciuto et al. (2019) discovered that there is little correlation between the rank found by weight sharing and rank of actual performance. Anonymous (2020) also conducted a benchmark experiment that shows a similar instability when the search space contains thousands of child models, by leveraging ground truth results measured by Ying et al. (2019). However on the other hand, research on transfer learning (Razavian et al., 2014), where a particular model trained on a particular task can work well on another task, and multitask learning (Luong et al., 2015), where multiple models trained for multiple tasks share the same weights during training, suggest otherwise and encourage the weights to be shared among child models, to reduce the long training time from scratch to convergence (Pham et al., 2018). Therefore, in this paper we show whether weight sharing helps and why, using comprehensive experiments.

## 3 WEIGHT-SHARING: VARIANCE AND INVARIANCE

### 3.1 METHODOLOGY

The space of a typical neural architecture search task usually has more than $10^{10}$ different child models (Tan et al., 2019; Liu et al., 2018a; 2019), thus, it is impossible to train them all, which leads to the problem that without ground truth it is hard to assess how good the found child model is in the search space. To solve this problem, we down-scale search space under the assumption that small search space is easier than large search space, if the search methods works in large search space they are also supposed to work in small search space.

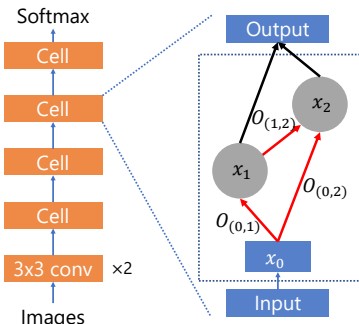

$O_{(i,j)}$ can be one of the following:

1. $3 \times 3$ max pooling
2. $3 \times 3$ separable convolutions
3. $5 \times 5$ separable convolutions
4. $3 \times 3$ dilated separable convolutions

Figure 1: Down-scaled search space.

Following DARTS (Liu et al., 2018c), we design a search space for a cell, as shown in Figure 1, and stack four cells each of which has the same chosen structure, forming a convolutional neural network. A cell is defined as a directed acyclic graph (DAG) of $n$ nodes (tensors) $x_1, \ldots, x_n$. A cell starts with $x_0$, which is the output tensor of its previous cell fed through a 1x1 conv layer to match the targeted number of channels in the current layer. The output of the cell is simply the sum of $x_1, \ldots, x_n$. The DAG is designed to be densely connected, *i.e.,*

$$x_j = \sum_{0 \leq i < j} O_{(i,j)}(x_i) \tag{1}$$

where $O_{(i,j)}$ is the selected operation at edge $(i,j)$. In the down-scaled search space, each cell contains only two nodes (*i.e.,* $n = 2$) and $O_{(i,j)}$ is one of the four primitive operations in Figure 1. Thus, a child model only has $4^3 = 64$ possible choices, which makes it easy to have ground truth. For convenience, we also name all the child models with three digits (each digit is in $[1, 4]$), denoting the choice of $O_{(0,1)}, O_{(0,2)}, O_{(1,2)}$ respectively.

As introduced in the previous section, sharing a single copy of weights can be seen as training an expanded super-net. To better understand the effect of weight sharing, we simplify the training process. Specifically, we uniformly generate child models. Each mini-batch trains one child model and only the weights of this model are updated by back-propagation. After training the shared weights for a number of epochs, we use these shared weights to evaluate the performance of all child models

on the validation set. On the other hand, the ground truth performance of each child model is obtained by training each of them independently from scratch with the same setting as weight sharing, and averaging over 10 runs with different random seeds for initializations. The lookup table can be found in Appendix C.

For the rest of the experiments listed in this paper, if not otherwise specified, the models are trained with the dataset of CIFAR-10 on an NVidia K80 GPU. We use SGD with momentum $0.9$ and weight decay $10^{-3}$ as our optimizer. The initial learning rate is set to $0.025$ and annealed down to $0.001$ following a cosine schedule without restart (Loshchilov & Hutter, 2016). The batch size is set to $256$. Number of epochs is $200$. Detailed experiment settings are described in Appendix A.

## 3.2 VARIANCE OF WEIGHT SHARING

To measure stability and performance of weight sharing methods, we first need to measure a rank, as weight sharing methods use the performance ranks of child models on validation set to choose the final output child model. We leverage Kendall's rank correlation coefficient, *i.e.,* Kendall's Tau (Kendall, 1938), which provides a measure of correspondence between two ranks $R_1$ and $R_2$. Intuitively, $\tau(R_1, R_2)$ can be as high as $1$ if $R_1$ and $R_2$ are perfectly matched, or as low as $-1$ when $R_1$ and $R_2$ are exactly inverted. We use *instance* to denote the procedure of training the super-net and generating a rank $R_i$ of child models on validation set. We then define the following three metrics.

- **S-Tau**: S-Tau is to measure the stability of generated ranks from multiple instances. For $N$ instances with ranks $R_1, R_2, \ldots, R_N$, S-Tau can be calculated as,

$$\frac{2}{N(N-1)} \sum_{1 \leq i < j \leq N} \tau(R_i, R_j) \tag{2}$$

- **GT-Tau**: This metric is to compare the rank produced by an instance with ground truth rank. We also use Kendall's Tau to measure the correlation of the two ranks, *i.e.,* $\tau(R, R_{\text{gt}})$.

- **Top-n-Rank (TnR)**: It is to measure how good an instance is at finding the top child model(s). TnR is obtained by choosing the top $n$ child models from the generated rank of an instance and finding the best ground truth rank of these $n$ child models.

Similar to a good deep learning model that could constantly converge to a point that has similar performance, weight-sharing NAS is also expected to have such stability. If we use the same initialization seed and the same sequence of child models for mini-batches in different instances, they will produce the same rank after the same number of epochs. To measure the stability when applying different seeds or sequences, we do several experiments and the results are shown in Table 1. For the first three rows, each of them is an experiment that runs $10$ instances. The first one makes initialization seed different in different instances while keeping other configurations the same. The second one uses a random child model sampler with different seeds to generate different order of the $64$ child models for different instances, each instance repeats the order in mini-batch training, and seeds for weight initializations are the same for those instances. The only difference between the second and the third one is that after every $64$ mini-batches a new order of the child models is randomly generated for the next $64$ mini-batches, we call it different order with shuffle.

From the numbers, we can see that different initialization seeds make the generated ranks very different. Some instances generate high correlation ranks while some others even show negative correlation. To give an intuitive understanding of the S-Tau values, we also show two baselines, *i.e.,* random rank which includes $10$ randomly generated ranks and ground truth which trains the $64$ child models independently and generate a rank in every instance. The rank generated by training child models independently is much more stable. S-Tau of different orders with or without shuffle is lower than $0.5$. But S-Tau values of the three experiments under the same epoch are not comparable, because S-Tau varies a lot in different epochs. For example, as shown in Figure 2, S-Tau of the $10$ instances with different seeds varies in the range of $0.4$ even in the last several epochs — it could be as low as $0.3$ or as high as $0.7$, which, to some extent, explains inconsistent results from previous works (Sciuto et al., 2019). **Observation 1:** The rank of child models on validation set is very unstable in different instances.

We also compared the generated ranks with the ground truth rank with GT-Tau as shown in Table 2. Similar to S-Tau, GT-Tau values of the three experiments are also much lower than that of ground

Table 1: Instability of multiple runs (*i.e.,* instances) measured with S-Tau. Max Tau means the maximum value of the $\frac{N(N-1)}{2}$ Taus. Similarly, Min Tau is the minimum value. The numbers are obtained at the 200-th epoch.

| Experiments | S-Tau | Max Tau | Min Tau |
|---|---|---|---|
| Different seeds | 0.5415 | 0.7977 | 0.2471 |
| Different orders | 0.3930 | 0.7021 | −0.0129 |
| Diff. orders (shuffle) | 0.4403 | 0.7163 | 0.0764 |
| Random rank | 0.0382 | 0.2181 | −0.1552 |
| Ground truth | 0.7120 | 0.8191 | 0.6650 |
| Different epochs | 0.5310 | 0.8752 | 0.0918 |

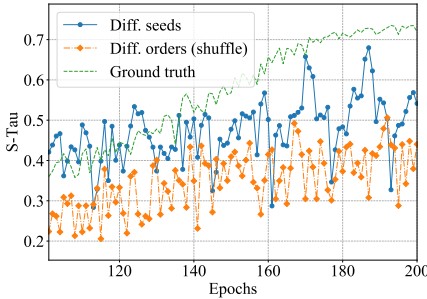

Figure 2: S-Tau evaluated after every epochs for "Diff. seeds", "Diff. orders (shuffle)" and "Ground truth".

Table 2: Comparison with ground truth with GT-Tau and TnR. Each number is an average of 10 numbers, either from 10 instances or from 10 epochs of one instance. The subscript shows the standard variance of these 10 numbers.

| | GT-Tau | T1R | T3R |
|---|---|---|---|
| Different seeds | $0.4567_{\pm 0.1478}$ | $18.5000_{\pm 1.2042}$ | $17.6000_{\pm 0.4899}$ |
| Different orders | $0.4625_{\pm 0.0993}$ | $16.9000_{\pm 5.7000}$ | $11.5000_{\pm 5.3712}$ |
| Diff. orders (shuffle) | $0.5108_{\pm 0.0665}$ | $15.6000_{\pm 8.4758}$ | $11.2000_{\pm 5.5462}$ |
| Ground truth | $0.7985_{\pm 0.0257}$ | $4.8000_{\pm 3.8419}$ | $1.6000_{\pm 0.9165}$ |
| Different epochs | $0.5053_{\pm 0.1399}$ | $15.2000_{\pm 6.9828}$ | $13.2000_{\pm 7.4404}$ |

truth, and the variance of GT-Tau across different instances is also high, which implies that the generated rank is not qualified to guide the choosing of good-performing child models. This is further proved by T1R and T3R. T1R ranges from 15 to 19, meaning that if choosing top 1 child model it is unlikely to obtain a good-performing model. T3R is slightly better than T1R, but at the cost of training more child models from scratch, which is usually not affordable for large search space. **Observation 2:** Though weight sharing shows the trend of following ground truth (has correlation), the generated rank is still far from the ground truth rank, seemingly having a hard limit.

Now that multiple instances have shown high variance, how about the stability of one single instance near the end of the training? We then look into a single instance by measuring variance of the ranks generated in consecutive epochs. Specifically, for each instances from the previous three experiments, we obtain 10 ranks each from one of the last 10 epochs (*i.e.,* 191 – 200), measure the stability of the 10 ranks and compare them with ground truth rank. We calculate S-Tau to show the mutual correlation among these 10 ranks. This value turns out to vary between 0.39 to 0.63 for different orders (shuffle), which means there is high variance between epochs even within a single instance. We show the median number among instances in Table 1. GT-Tau also varies a lot along epochs. Taking one instance from "Diff. orders (shuffle)" with final GT-Tau 0.47, we found that, as shown in Table 2, actually its GT-Tau varies between 0.1 to 0.7, with standard variance 0.14, in the last 10 epochs. **Observation 3:** The generated ranks in the last several epochs of the same instance are highly unstable, indicating that picking a different epoch to generate the rank has great impact on the finally obtained performance.

### 3.3 EXPLOITABLE FROM VARIANCE

Though the generated ranks show high variance, there is some statistical information that can be extracted from the variance. For the "Diff. orders (shuffle)" experiment, we have 10 ranks on the 200th epoch of the 10 instances. For each child model, we retrieve its rank values in the 10 ranks, and show the distributions in Figure 3a. The child models are ordered with their ground truth accuracy, the left ones are better than the right ones. We can see that bad-performing models are more likely ranked as bad ones (also observed by Bender et al. (2018)), while it is almost not possible to tell which one is better from the good-performing ones. Furthermore, we evaluate the ranks generated

from the last 10 epochs of the same instance in the same way. The result is shown in Figure 3b, which is a almost same result as the multi-instance experiment, implying that we can simply run one instance and generate multiple ranks from different epochs, these ranks can be used to quickly filter out bad-performing models. **Insight 1:** Though weight sharing is unstable, the generated ranks can be leveraged to quickly filter out bad-performing child models, and potentially used to do search space pruning, *e.g.,* progressively discarding the bottom-ranked child models and only further training the top ones.

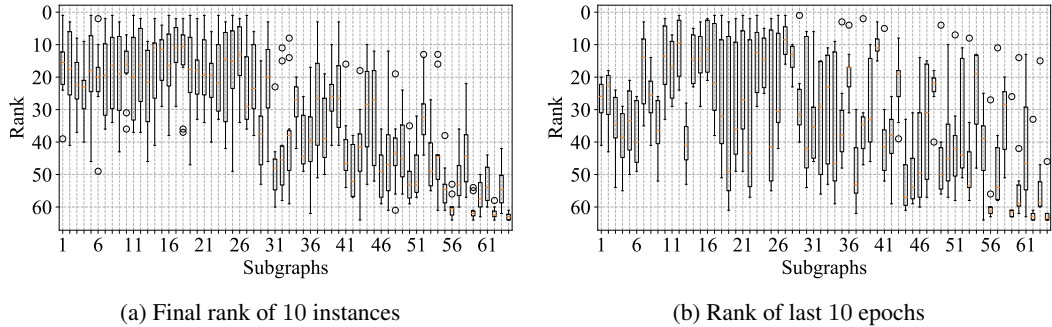

(a) Final rank of 10 instances

(b) Rank of last 10 epochs

Figure 3: Distribution of rank achieved for each child model, ordered from the ground-truth-best to worst. Each box extends from the lower to upper quartile values of its corresponding data, with a line marking the median. The whiskers show the range of the data. Outliers are marked with circles.

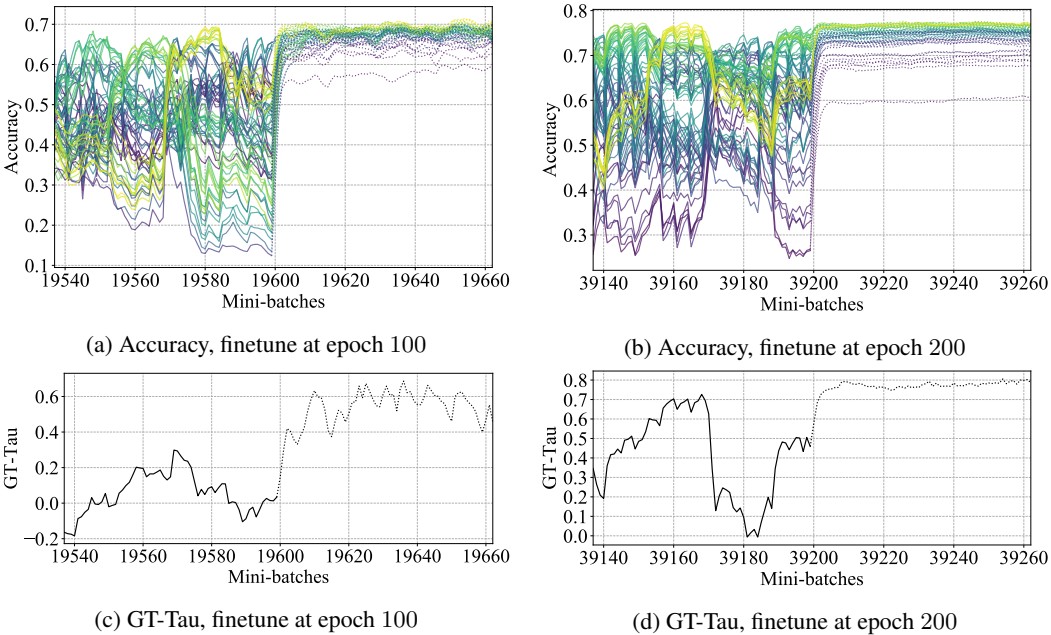

(a) Accuracy, finetune at epoch 100

(b) Accuracy, finetune at epoch 200

(c) GT-Tau, finetune at epoch 100

(d) GT-Tau, finetune at epoch 200

Figure 4: For the solid line part, all the child models share one copy, while the dotted lines represent the part where each child model training independently. The accuracies are evaluated after each mini-batch, and GT-Tau is calculated correspondingly.

As shown in the previous experiment, a single instance can converge to a state that the performance of some child models can no longer be distinguished, which can be seen as a kind of stable state. In this state, further training of the super-net does not improve the quality of ranks (also proved in § 4.1). We propose to finetune each child models independently by inheriting the weights from a snapshot of the super-net. Specifically, in Figure 4a, we train the super-net for 100 epochs and then finetune the child models for 64 mini-batches. We can see from Figure 4c that GT-Tau quickly increases up to 0.6 with only 10 mini-batches. Finetuning from the 200-th epoch shows even better results in Figure 4b:

the convergence is faster (using 5 mini-batches) and GT-Tau is more stable (close to $0.8$, Figure 4d). **Insight 2:** Weight sharing super-net could act as a good pretrained model. Finetuning child models with limited mini-batches could greatly improve the quality of the rank.

# 4 Understanding Variance of Weight Sharing

Understanding the source of variance is the key to better leverage the power of weight sharing. In this section, we measure more numbers and design different experiments to understand where the variance comes from and how to reduce the variance.

## 4.1 Source of Variance

The first step is to find out the reasons why there is high variance in consecutive epochs of a single instance. We pick an instance from the "Diff. seeds" experiment. In this instance we evaluate the performance of the 64 child models on the validation set after *every mini-batch* near the end of training. The result is shown in Figure 5a. The curves has obvious periodicity with the length of 64 mini-batches, *i.e.,* the number of child models. Curves with light colors are mainly located at the upper of the figure, but they are not always the better ones. In some mini-batches the curves with darker colors perform better. In Figure 5, if the $i$-th mini-batch trains child model $c$, we use a diamond marker to label $c$'s curve. We can see that in most of mini-batches training a child model makes this child model performs the best. Some bad-performing child models can also become the best one by training them in mini-batches. It implies that training a child model can easily perturb the rank of the previous mini-batch.

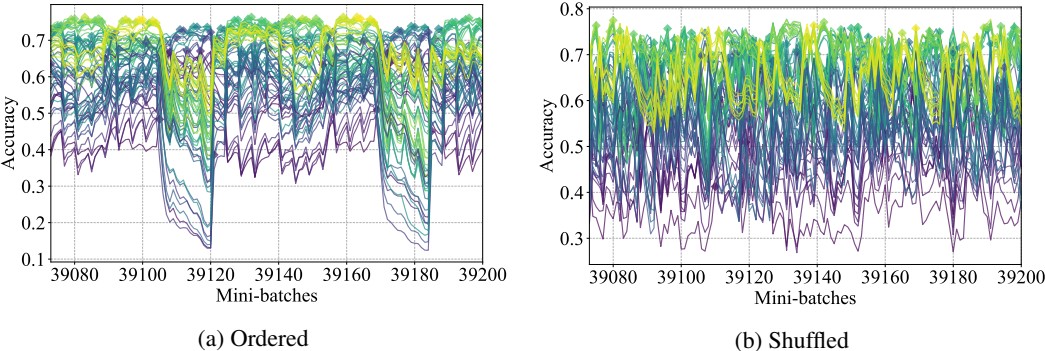

(a) Ordered                                             (b) Shuffled

Figure 5: The validation performance of all child models, evaluated after each of the last 128 mini-batches. Each curve corresponds to one child model. Markers are marked on the child model trained in the current mini-batch. Lighter colors correspond to higher ground truth ranks. The difference between ordered and shuffled is "shuffled" generate a new order of child models every 64 mini-batches. Each figure shows a window of 128 mini-batches. It's clear that in Figure 5a a pattern is repeated twice and the periodicity is 64 mini-batches.

To further verify this phenomenon, we show the result of an instance with shuffled sequence of child models in Figure 5b. There is no periodicity, but other results are very similar. Though curves with light colors generally perform better than the other curves, it is still hard to tell which one of them is better than others. The instability of rank during the last mini-batches in Figure 5 also implies the instability of GT-Tau, which means GT-Tau retrieved at the end of the training can also be highly unreliable. In fact, for the instance shown in Figure 5a, GT-Tau varies between $0.0$ and $0.6$ in the last 128 mini-batches.

In the rest of this section, we decrease the degree of weight sharing with different approaches. To faithfully reveal their effectiveness, we calculate the average GT-Tau for an instance in the last $k$ mini-batches due to the variance among mini-batches. We call it *GT-Tau-Mean-k*. To see the stability of GT-Tau, we also obtain the standard variance of these $k$ GT-Taus, which we call *GT-Tau-Std-k*.

Table 3: GT-Tau-Mean-64 and GT-Tau-Std-64, averaged over 10 instances. The subscript is the standard variance corresponding to the average.

(a) Group Sharing

| | Grouping By Random | | Grouping By Similarity | |
| --- | --- | --- | --- | --- |
| $m$ | Mean-64 | Std-64 | Mean-64 | Std-64 |
| 1 | $0.4988_{\pm 0.0320}$ | $0.1497_{\pm 0.0455}$ | $0.4988_{\pm 0.0320}$ | $0.1497_{\pm 0.0455}$ |
| 2 | $0.4577_{\pm 0.0424}$ | $0.1371_{\pm 0.0233}$ | $0.3425_{\pm 0.0490}$ | $0.1442_{\pm 0.0424}$ |
| 4 | $0.2736_{\pm 0.0216}$ | $0.1340_{\pm 0.0235}$ | $0.7075_{\pm 0.0137}$ | $0.0702_{\pm 0.0156}$ |
| 8 | $0.2539_{\pm 0.0463}$ | $0.1462_{\pm 0.0172}$ | $0.6917_{\pm 0.0267}$ | $0.0457_{\pm 0.0088}$ |
| 16 | $0.1658_{\pm 0.0316}$ | $0.1255_{\pm 0.0155}$ | $0.7200_{\pm 0.0213}$ | $0.0411_{\pm 0.0098}$ |
| 32 | $0.2903_{\pm 0.0256}$ | $0.1104_{\pm 0.0087}$ | $0.7360_{\pm 0.0164}$ | $0.0364_{\pm 0.0096}$ |
| 64 | $0.8032_{\pm 0.0255}$ | $0.0151_{\pm 0.0036}$ | $0.8032_{\pm 0.0255}$ | $0.0151_{\pm 0.0036}$ |

(b) Prefix Sharing

| $k$ | Mean-64 | Std-64 |
| --- | --- | --- |
| 0 | $0.6960_{\pm 0.0193}$ | $0.0129_{\pm 0.0066}$ |
| 1 | $0.4837_{\pm 0.0822}$ | $0.0939_{\pm 0.0545}$ |
| 2 | $0.4159_{\pm 0.0504}$ | $0.1925_{\pm 0.0362}$ |
| 3 | $0.4448_{\pm 0.0689}$ | $0.1881_{\pm 0.0468}$ |
| 4 | $0.5174_{\pm 0.0340}$ | $0.1592_{\pm 0.0163}$ |

## 4.2 GROUP SHARING: REDUCE VARIANCE

### 4.2.1 GROUPING BY RANDOM

Trying to reduce variance, we divide the child models into groups, each of which is trained independently and maintains one copy of weights. We first, naively, randomly divide all the child models in a search space into $m$ groups. Therefore, for a search space of $n$ child models, $m = 1$ corresponds to fully weight sharing and $m = n$ corresponds to no weight sharing.

We conduct experiments on our search space of 64 child models. $m$ is chosen from 1, 2, 4, 8, 16, 32 and 64. For each $m$, we repeat the experiment for 10 instances, with the same group partition, but different seeds for initialization of weights. We run each group for 200 epochs and evaluate the validation accuracy of every child model at each of the last 64 mini-batches to obtain GT-Tau-Mean-64, GT-Tau-Std-64, and average them over instances, as shown in Table 3a.

Actually, breaking down the complexity through random grouping does not increase stability but actually backfires. From $m = 16$, the worst performing case, we take an instance for case study. We calculate GT-Tau-Mean-64 for each group, *i.e.,* the including child models. The average GT-Tau-Mean-64 of the 16 groups is as low as $0.2570$. To compare, We partition the ranks generated by an instance from $m = 1$ into those 16 groups, and calculate GT-Tau-Mean-64 for each group in the same way, the average GT-Tau-Mean-64 is $0.5610$ which is much higher than $0.2570$. Thus, we argue that the downgrading of GT-Tau on the full rank mainly comes from internal instability inside groups. By examining the accuracy and rank of child models in each group, we find that interference among child models commonly exists in almost all the groups, even for $m = 32$ where there are only 2 child models per group. Such interference causes a drastic reordering of the rank of child models.

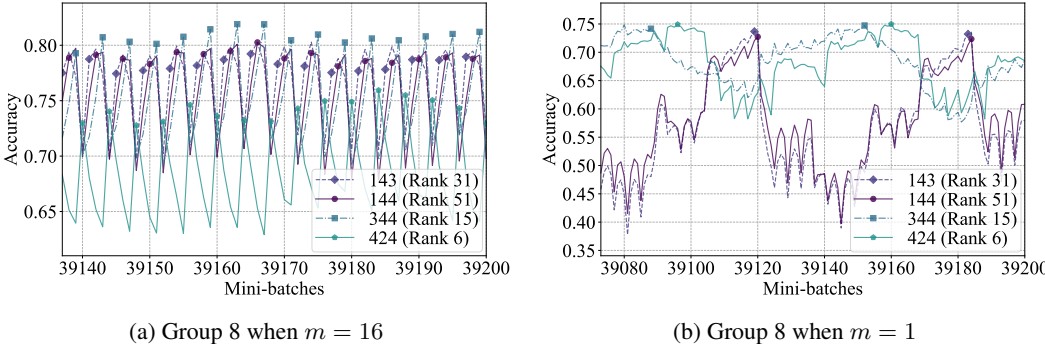

(a) Group 8 when $m = 16$           (b) Group 8 when $m = 1$

Figure 6: Validation accuracy of a group with 4 child models. Markers are marked on child models trained in the current mini-batch.

To dive deeper, we plot the accuracy of the child models from a group in the instance of $m = 16$, as shown in Figure 6a. Some child models facilitate each other, while some others degrade each other. Specifically, the child models 143, 144 and 344 go up and down consistently, while the child model

424 acts exactly the opposite. Note that, 424 is the best-performing one in ground truth but performs the worst in this group, which indicates that *properly choosing the child models for co-training (i.e., weight sharing) is the key to obtain a good rank*. This is further supported by Figure 6b which shows the accuracy of these four child models when $m = 1$. With more other child models joining in for co-training, the four child models' curves become very different from that in Figure 6a. For example, the curves of 344 and 424 become very similar.

On the other hand, from the first column in Table 3a, we can see that GT-Tau-Mean-64 first decreases then increases when $m$ changes from 1 to 64. A possible explanation is that when many child models share a single copy of weights, a single child model cannot bias the group a lot, while when each group becomes very small, the child models share less weights with each other, thus also not easy to bias each other too much. **Observation 1:** Two child models have (higher or lower) interference with each other when they share weights. A child model's validation accuracy highly depends on the child models it is jointly trained with.

### 4.2.2 GROUPING BY SIMILARITY

According to the observations above, we further explore how it works by grouping child models based on similarity. We sort the child models lexicographically from 111 to 444, then slice the sequence into $m$ groups. For example, when $m = 8$, group 1 has the child models from 111 to 124, group 2 is from 131 to 144, group 3 is from 211 to 224, and so on. The results are shown on the right of Table 3a. We can see that there is a global trend of stabilization from $m = 1$ to 64, *i.e.,* GT-Tau-Mean-64 goes higher and GT-Tau-Std-64 gets lower. **Observation 2:** A *smart* grouping can generally improve the stability of training.

In our case, "smart" means "similar". However, this might not be the case for more complex search space, where "similar" can be poorly defined, or the range of the space is too large, or even infinite. Admittedly, for larger and more complex search space, such smart grouping has to be found by other means, *e.g.,* correlation matrix among child models. We leave it in future work.

### 4.3 PREFIX SHARING

Inspired by the great success of transfer learning (Caruana, 1995; Mesnil et al., 2011; Kornblith et al., 2019), we try to do a similar thing by sharing one copy of "backbone" network while keeping a separated copy of "head" for each child model. In particular, we use $k$ to denote the number of cells shared. When $k = 0$, only the first two conv layers are shared. When $k = 4$, all the layers except the final fully-connected layers are shared. In the experiments, we increase the total epochs from 200 to 2000, as the models require more computation to reasonably converge.

The results are shown in Table 3b. Obviously, sharing fewer cells improves the GT-Tau and accuracy (more experiment numbers can be seen in Appendix B). The performance becomes better but at the cost of consuming more computation power. For example, though a high and stable GT-Tau is obtained when $k = 0$, it takes over 1000 epochs for it to climb up to above 0.6. But still, this cost is much lower than ground truth, which takes $64 \times 200 = 12800$ epochs in total.

## 5 CONCLUSION

Neural architecture search is becoming a feasible way to explore new models, but its excessive computation cost impels researchers to resort to the power of weight sharing. In this paper we use comprehensive experiments to have a close look at weight sharing, and illustrate many interesting insights. By designing more sophisticated experiments, we further dig out the reasons of high variance of weight sharing. To further improve NAS, we believe the key is to figure out how to smartly leverage shared weights. This paper suggests controlling the degree of weight-sharing, either model-based and rule-based, evaluating them on the small search space and providing deeper insights. We hope to inspire the community to find more stable yet efficient approaches.

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

## A  EXPERIMENT SETTINGS

### A.1  OVERVIEW

Since our network architecture is very similar with that introduced in DARTS Liu et al. (2018c), we basically follow the settings in DARTS, with a few modifications.

1. **Batch size:** We following the settings from a PyTorch implementation of DARTS[2] and set batch size to 256. The current batch size divides one epoch into 196 mini-batches.

2. **Number of epochs:** Previous works Liu et al. (2018c); Li & Talwalkar (2019) all set the number of epochs as a constant number. We follow the similar settings and set it to 200. Empirically, we experiment with epochs from 50 to 2000, though they show different final accuracies, all our Tau's reported in this paper seems insensitive to the number of epochs trained, except when the number of epochs is too small for the instance to reach a plateau. See Appendix A.2.1 and Appendix B for more information.

3. **Learning rate:** Following DARTS, we use an initial learning rate 0.025, annealing down to 0.001 following a cosine schedule depending on the total number of epochs.

4. **Optimizer:** We use SGD as our optimizer to train graphs. Weight decay is set to be $10^{-3}$ as smaller networks often needs larger weight decays. The momentum is set to be 0.9, and the velocities (momentum buffer) for parameters shares if the corresponding parameter shares.

5. **Network:** The first 2 conv layers of the entire network expand the 3-channel image into 48 channels. Each of the following 4 cells will first compress the input into 16 channels and feeds into the DAG and concat the output of nodes (32 channels in our settings). The momentum of batch normalization is set to be 0.4 (under the semantics of PyTorch[3]).

For reproducibility, we set a certain seed before running all the experiments, and we removed the non-deterministic behavior in CuDNN.

### A.2  EFFECTS OF HYPERPAMETERS

In order to justify results of this paper are not just any coincidence of a carefully picked set of hyperparameters. We compare some of the results here with those with difference choices.

### A.2.1  NUMBER OF EPOCHS

To show that the phenomenon in Figure 5a is not just a result of training too few epochs, we repeat the same experiment with the same experiment settings, with number of epochs set to 2000, obtaining Figure 7, which shows similar periodicity and instability, despite the overall accuracy is higher.

### A.2.2  MOMENTUMS

Larger momentums help preserve the information from previous mini-batches, during which other child models are training, thus, presumably stabilize the training.

**Batch norm**  Following the definitions in PyTorch, higher BN momentum indicates that the mean and variance in the current mini-batch have a higher weight. We compare accuracy curves for BN momentum is lower (0.1) and higher (0.9), each repeating 3 times with different seeds for initializations. Experiments show that lower BN momentum helps stabilize the training in a short term, but it's still trembling in the long term, see Figure 8.

**SGD momentum**  We compare the results of accuracy curves when SGD momentum is set to 0 with 0.9. Results are shown as in Figure 9. The accuracy seems to vary in a greater range, compared to Figure 5a, and GT-Tau varies between $-0.1$ and $0.5$, which is more unstable.

---

[2]`https://github.com/khanrc/pt.darts`

[3]The momentum here is different from that used in optimizers. Mathematically, it's $\hat{x}_{new} = (1 - \text{momentum}) \cdot \hat{x} + \text{momentum} \cdot x_t$. Reference: `https://pytorch.org/docs/stable/_modules/torch/nn/modules/batchnorm.html`

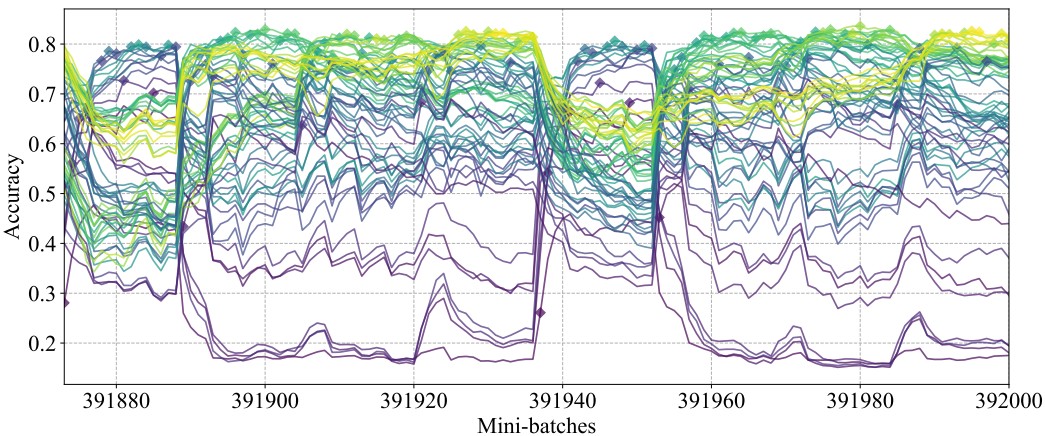

Figure 7: Validation accuracy for all child models for each of the last 128 mini-batches of 2000 epochs.

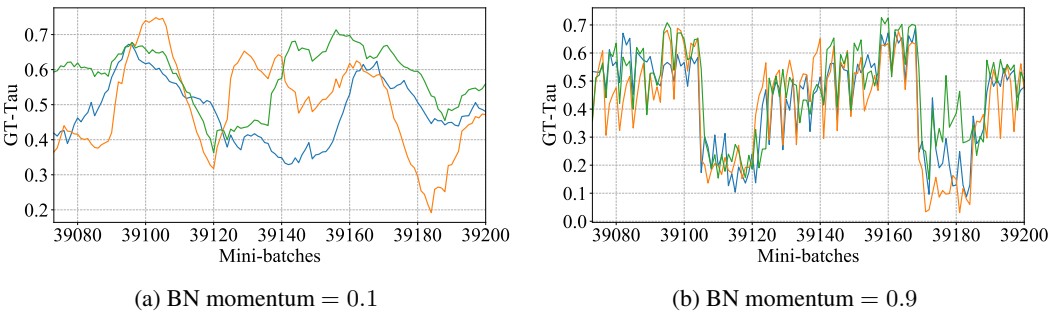

(a) BN momentum = 0.1                  (b) BN momentum = 0.9

Figure 8: GT-Tau curves over last 128 mini-batches. Each color represents one instance.

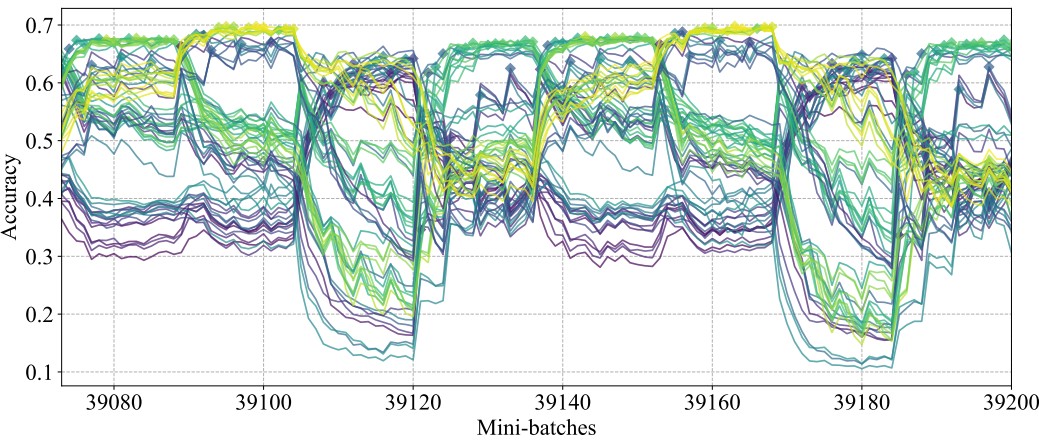

Figure 9: Validation accuracy for all child models for each of the last 128 mini-batches, when SGD momentum is set to 0.

## B    PREFIX SHARING: DETAILS

Table 4: Supplementary to Table 3. GT-Tau-Mean-64 and GT-Tau-Std-64 follow the same scheme. Similar to these two metrics, accuracy of each child model in each instance is first averaged over $64$ mini-batches, and then mean and std of all child model accuracies in an instance are calculated. Finally we average mean and std from each instance over $10$ instances. The subscript is the standard variance of the corresponding average.

| Epochs | $k$ | GT-Tau-Mean-64 | GT-Tau-Std-64 | Accuracy Mean | Accuracy Std |
|--------|-----|----------------|---------------|---------------|--------------|
| 2000 | 0 | $0.6960_{\pm 0.0193}$ | $0.0129_{\pm 0.0066}$ | $0.7793_{\pm 0.0036}$ | $0.0245_{\pm 0.0010}$ |
| 2000 | 1 | $0.4837_{\pm 0.0822}$ | $0.0939_{\pm 0.0545}$ | $0.7691_{\pm 0.0089}$ | $0.0378_{\pm 0.0040}$ |
| 2000 | 2 | $0.4159_{\pm 0.0504}$ | $0.1925_{\pm 0.0362}$ | $0.6972_{\pm 0.0159}$ | $0.0640_{\pm 0.0081}$ |
| 2000 | 3 | $0.4448_{\pm 0.0689}$ | $0.1881_{\pm 0.0468}$ | $0.6636_{\pm 0.0178}$ | $0.0823_{\pm 0.0113}$ |
| 2000 | 4 | $0.5174_{\pm 0.0340}$ | $0.1592_{\pm 0.0163}$ | $0.6639_{\pm 0.0166}$ | $0.0995_{\pm 0.0166}$ |
| 200 | 0 | $0.0441_{\pm 0.1348}$ | $0.0145_{\pm 0.0044}$ | $0.6083_{\pm 0.0065}$ | $0.0173_{\pm 0.0020}$ |
| 200 | 1 | $0.3895_{\pm 0.0731}$ | $0.0989_{\pm 0.0339}$ | $0.6435_{\pm 0.0114}$ | $0.0298_{\pm 0.0058}$ |
| 200 | 2 | $0.4104_{\pm 0.0496}$ | $0.1801_{\pm 0.0418}$ | $0.6035_{\pm 0.0138}$ | $0.0535_{\pm 0.0105}$ |
| 200 | 3 | $0.4545_{\pm 0.0581}$ | $0.1570_{\pm 0.0385}$ | $0.6081_{\pm 0.0181}$ | $0.0667_{\pm 0.0086}$ |
| 200 | 4 | $0.4899_{\pm 0.0501}$ | $0.1622_{\pm 0.0534}$ | $0.6177_{\pm 0.0171}$ | $0.0800_{\pm 0.0181}$ |

As shown in Table 4, in the case of 200 epochs, accuracy is much lower than that after $2000$ epochs. Also, GT-Tau shows an ascending trend, which is quite the opposite of the phenomenon observed in larger epochs. We argue that this is due to the insufficient training for smaller $k$. Prefix sharing causes the networks to having multiple copies and these copies cannot be trained independently (as opposed to parallelizing the groups in §4.2). Running 2000 epochs in prefix sharing is fair and necessary, with acceptable time cost (about 2 days on a GPU for one instance).

## C    GROUND TRUTH LOOKUP TABLE

Shown in Table 5 is a table containing accuracies of all $64$ child models, along with their ranks.

Table 5: Accuracies of child models when they are trained independently. The three digits of child model labels correspond to $O_{(0,1)}$, $O_{(0,2)}$ and $O_{(1,2)}$ respectively. Evaluations are repeated 10 times with different seeds of weight initialization, and averages and standard variances are calculated. Ranks are based on the average values. The higher, the better.

| Child Model | Acc Mean | Acc Std | Rank | Child Model | Acc Mean | Acc Std | Rank |
|---|---|---|---|---|---|---|---|
| 111 | 70.13 | 0.23 | 64 | 311 | 81.78 | 0.20 | 53 |
| 112 | 78.23 | 0.75 | 62 | 312 | 83.52 | 0.25 | 27 |
| 113 | 81.06 | 0.42 | 57 | 313 | 83.31 | 0.34 | 32 |
| 114 | 79.98 | 0.40 | 59 | 314 | 83.48 | 0.26 | 28 |
| 121 | 77.77 | 0.37 | 63 | 321 | 82.91 | 0.25 | 41 |
| 122 | 81.77 | 0.50 | 54 | 322 | 84.24 | 0.22 | 18 |
| 123 | 83.10 | 0.21 | 39 | 323 | 84.33 | 0.29 | 13 |
| 124 | 82.58 | 0.43 | 49 | 324 | 84.62 | 0.19 | 11 |
| 131 | 81.14 | 0.21 | 56 | 331 | 82.91 | 0.22 | 42 |
| 132 | 83.15 | 0.36 | 38 | 332 | 83.98 | 0.13 | 25 |
| 133 | 83.21 | 0.28 | 34 | 333 | 84.70 | 0.53 | 10 |
| 134 | 82.77 | 0.19 | 48 | 334 | 84.47 | 0.41 | 12 |
| 141 | 79.62 | 1.04 | 60 | 341 | 83.39 | 0.18 | 29 |
| 142 | 82.86 | 0.17 | 46 | 342 | 84.32 | 0.51 | 14 |
| 143 | 82.87 | 0.31 | 43 | 343 | 84.23 | 0.35 | 19 |
| 144 | 82.07 | 0.38 | 51 | 344 | 84.32 | 0.49 | 15 |
| 211 | 78.67 | 0.49 | 61 | 411 | 80.43 | 0.73 | 58 |
| 212 | 81.86 | 0.37 | 52 | 412 | 82.94 | 0.44 | 40 |
| 213 | 83.31 | 0.28 | 31 | 413 | 82.87 | 0.19 | 43 |
| 214 | 82.81 | 0.30 | 47 | 414 | 83.21 | 0.53 | 34 |
| 221 | 81.42 | 0.15 | 55 | 421 | 83.27 | 0.21 | 33 |
| 222 | 83.20 | 0.30 | 36 | 422 | 84.76 | 0.56 | 8 |
| 223 | 84.22 | 0.31 | 20 | 423 | 84.73 | 0.33 | 9 |
| 224 | 84.17 | 0.32 | 22 | 424 | 84.77 | 0.57 | 7 |
| 231 | 82.87 | 0.44 | 45 | 431 | 83.39 | 0.25 | 30 |
| 232 | 83.81 | 0.43 | 26 | 432 | 84.80 | 0.42 | 6 |
| 233 | 84.28 | 0.28 | 16 | 433 | 84.84 | 0.57 | 5 |
| 234 | 84.19 | 0.50 | 21 | 434 | 85.23 | 0.17 | 1 |
| 241 | 83.20 | 0.31 | 37 | 441 | 82.19 | 0.26 | 50 |
| 242 | 84.28 | 0.14 | 17 | 442 | 84.87 | 0.59 | 3 |
| 243 | 84.03 | 0.19 | 24 | 443 | 84.93 | 0.23 | 2 |
| 244 | 84.08 | 0.36 | 23 | 444 | 84.85 | 0.35 | 4 |

