# OpenReview forum: "Deeper Insights into Weight Sharing in Neural Architecture Search"
_ICLR.cc/2020/Conference — Reject_

### Official Review · AnonReviewer2 · 2019-10-09
**Official Blind Review #2**

**Rating:** 3

**Review:**

Many NAS methods rely on weight sharing. Notably in ENAS, a single weight tensor is used for all candidate operations each edge of a cell. In this paper, the authors take a small NAS search space (64 possible networks) and train each network separately to obtain their individual rankings. They then examine how this ranking correlates when  the same network is trained as part of a super-net with weight sharing, as in NAS algorithms.

Given the prevalence of NAS algorithms, an examination of the potential pitfalls of weight sharing is very important and I commend the authors for that. There are a lack of typos and grammatical errors, which is nice too!

I have two major issues with this paper however. Firstly, the scope is limited; everything is based on looking at 64 convnets trained on CIFAR-10, this makes it tricky to make any broad statements about weight-sharing in NAS. Secondly, the paper reads as a string of observations, and it is not clear what the takeaways are (it is hinted that one could reduce the search space for Figure 3, but this is not expanded on).

A few chronological comments:

- "Population based algorithm () is another popular approach" ---> "Population-based algorithms are another popular approach"

- "In this paper we try to answer" --> A little confidence wouldn't hurt :)

- "Surprisingly". I don't like the reader being told what is surprising/interesting etc. but maybe that's just me.

- As mentioned above, more detail on search space pruning would be really nice. For instance, are particular operations e.g. conv5x5 neglected.

- To clarify, when you have shared weights, but one candidate operation uses more parameters than another, do you just read off the weight tensor until you hit length? e.g. if convA uses X params, and convB uses Y params, are the parameters for ConvA weight(1:X) and convB weight(1:Y)?

- Literature review is good. Figure 1 is nice and straightforward.

- The figure and table captions could do with some more detail. For instance, in Figure 2 the caption should contain the take-home point of the figure.

- "there are some statistic information" --> "there is some statistical information"

- Figure 3 is nice. It looks like you can't tell what's good, but you can tell what's bad. A comparison of what architectures good v bad comprise off would be a nice addition.

- Figure 5 confused me, as there is a lot going on. What do ordered and shuffled mean? Is it just whether you are mixing up your minibatch selections? "The curve has obvious periodicity with the length of 64 mini-batches i.e. the number of child model" doesn't make sense to me. Could you elaborate?

- The accuracies in the plots look very low. ~80% for CIFAR-10 is really bad. Am I missing something?

Pros
------
- Good topic with a few interesting observations
- Relatively well-written

Cons
-------
- Very limited scope. Only 1 dataset and only 64 models
- The narrative is lacking, what are the key points that people using NAS should be aware of?

I recommend a weak rejection for this paper. The topic is interesting, but I haven't been convinced through the limited scope of the experiments, or the arguments made what the real point is. Should I stop weight sharing with NAS? Should I prune my search space? etc. A few neat observations is nice, but there is a lack of cohesion.

**Experience Assessment:**

I have read many papers in this area.

**Review Assessment: Checking Correctness Of Derivations And Theory:**

N/A

**Review Assessment: Checking Correctness Of Experiments:**

I assessed the sensibility of the experiments.

**Review Assessment: Thoroughness In Paper Reading:**

I read the paper at least twice and used my best judgement in assessing the paper.

---

> ### Author Response · Authors · 2019-11-09
> **Response to Official Blind Review #2**
>
> Thank you for your detailed review. The limited scope is selected based on the assumption that small search space is easier for NAS than a bigger one. If you ever question about that, we could do some extra experiments to make that up. However, I believe in this paper recently submitted (https://openreview.net/forum?id=SJx9ngStPH), they have already discovered a similar instability in Section 5.2.
>
> This paper is not meant to draw any conclusion, but to inspire future NAS researchers of things they could try. I agree that we should make a list of everything we could think of in our conclusion section.
>
> Here are chronological responses to your chronological comments.
>
> - Thank you for pointing out all the grammar mistakes and literature issues. Will fix in our revision.
>
> - About search space, a basic idea we propose here is to reduce the search space, discarding the bottom-ranked child models and only train the top-half ones with weight sharing. Do this over and over again until we find the top ones and train from scratch. The method is only a proposal, requiring more experiments.
>
> - Sharing conv weights follow the search space design of DARTS. We don't share a superkernel among different types of convolutions.
>
> - "what architectures good v bad comprise off" doesn't make sense to me. Are you interested in what architectures they are from ground-truth-best to worst, i.e., the ground truth ranking of 64 architectures? We will attach a table of all ground truth accuracies and ranks in Appendix.
>
> - The meaning of "ordered" and "shuffled" corresponds to "different seeds" and "different orders (shuffle)" in Table 1. Figure 5(a) shows a window of 128 mini-batches. The pattern is repeated twice and the periodicity is 64 mini-batches. Will make this clear in our revision.
>
> - The low accuracies shown in this paper are due to the bad architecture design. To simplify the network structure, we have removed the reduction cells. We use fewer cells and fewer nodes than usual. The networks are designed to be computationally efficient.

---

### Official Review · AnonReviewer1 · 2019-10-20
**Official Blind Review #1**

**Rating:** 3

**Review:**

First of all, I have to state that this is not my area of expertise. So, my review here is an educated guess.

The paper is an empirical study that looks into the effect of weight sharing in neural network architecture search. The basic idea is that by sharing weights among multiple candidate architectures, the training process can be significantly improved.  In the literature, there have been mixed results, either in favor of or against weight sharing. The question this paper aims to address is to determine if weight sharing is justifiable and to what extent.

The primary subject investigated by the authors is to determine if the variance of the ranks generated by different runs of the algorithm are highly correlated with each other (e.g. using Kendall rank correlation score). Then, they compared such results with the ground truth (i.e. every child is trained independently). They found that the ranks generated by weight sharing are indeed highly correlated with each other, but there is much larger variance in the ranks when compared to the ground truth method. To understand why this is non-trivial: (1) on one hand, weight sharing speeds up the training process by providing an initial point close to a local minimum, but (2) the local minimum point may or may not  be good for the new architecture.  Hence, one does not know apriori under what conditions would weight sharing be a good strategy.

The authors also looked into variance of the rank within the same instance by examining how the rank changes with mini-batch epochs. They found that variance is large even within the same instance.

My primary concern is that the paper is entirely empirical with little if any justification of the results. In addition, it is based on a single architecture and a single dataset. This would have been fine if the results were supported with explanation or theoretical justification. Second, the ultimate goal is to improve the prediction accuracy, not the ranking accuracy. These are not necessarily equivalent. For instance, it is possible that the ranks have a high variance simply because many of the candidate architectures have nearly equivalent performance so the order within them becomes nearly random (and unimportant). In fact, I think the results support this conclusion (see for example Figure 10). Third, some of the highlighted observations are trivial. For example, Observation 1, which states that "Two child models have (higher or lower) interference with each other when they share weights. A child model’s validation accuracy highly depends on the child models it is jointly trained with." I think this observation is trivial.

Some other comments:
- I would appreciate it if the authors could explain briefly how "prefix sharing" works so that the paper is self-contained.
- The goal is to help improve the speed of neural architecture search. The authors mention "hints for designing more efficient weight-sharing." Please state those conclusions precisely and clearly. I understand that the authors suggest similarity-based grouping. So, please mention clearly what you recommend in the conclusion section.


==========================
#post rebuttal remarks

Thanks for the response. Figure 10 was a typo from my end and I apologize for it. I actually meant figure 3.

As I said in my review, having an empirical study is acceptable provided that it covers many datasets, not just a single one.

**Experience Assessment:**

I do not know much about this area.

**Review Assessment: Checking Correctness Of Derivations And Theory:**

N/A

**Review Assessment: Checking Correctness Of Experiments:**

I assessed the sensibility of the experiments.

**Review Assessment: Thoroughness In Paper Reading:**

N/A

---

> ### Author Response · Authors · 2019-11-09
> **Response to Official Blind Review #1**
>
> Thank you for your detailed review.
>
> Lack of theoretical foundation is actually an open problem in weight-sharing NAS. This paper is an empirical study of weight sharing and it's not meant to be theoretical. We've also included some explanation to phenomenons in our paper, but most of them are just hypothesis.
>
> We don't agree that distinguishing performance-nearly-equivalent architectures is unimportant. If there were to be many architectures sharing the "best accuracy", one could just do random search: they don't need all those fancy NAS algorithms. It's true that we may not care about the overall rank, but only the top ones. However, as shown Table 2, even the top ones have a very bad ranking of more than 10 in average: it's on average worse than 1/4 of the child models in search space. In a large search space, it will become a candidate you will never try to train from scratch.
>
> Also, I'm sure we don't have a Figure 10 in our paper.
>
> This paper is supposed to inspire NAS researchers in future research, therefore we provide all these experiments and possible things to try. We agree that writing a list in the conclusion section might be better.
>
> We will elaborate the method of prefix sharing in our revision.

---

### Official Review · AnonReviewer3 · 2019-10-24
**Official Blind Review #3**

**Rating:** 3

**Review:**

This paper studies weight sharing in neural architecture search (NAS). It constructs a mini search space with 64 possible choices, and performs various comparisons and studies in an exhaustive way. Some of the observations are quite interesting, exploring the limitations of weight sharing.

My biggest concern is the limited search space. Unlike other NAS works that usually have search space size > 10^10, this paper focuses on a very small search space (64 options in total). Because the search space is so small, a small change in any search option might cause a big difference for the sampled model, which possibly lead to some of the instability observed in this paper (such as observation 3 in Section 3.2 and the implication "training a child model can easily perturb the rank of the previous mini-batch in section 4.1). However, this might not be true if the search space is big, where changing a few search options may not affect the supernet significantly.

It would be great if the authors can perform similar study on a larger search space. If evaluation for large search space is difficult, you may consider some pre-defined accuracy lookup tables (such as NAS-Bench-101: https://arxiv.org/abs/1902.09635).


**Experience Assessment:**

I have published in this field for several years.

**Review Assessment: Checking Correctness Of Derivations And Theory:**

I assessed the sensibility of the derivations and theory.

**Review Assessment: Checking Correctness Of Experiments:**

I assessed the sensibility of the experiments.

**Review Assessment: Thoroughness In Paper Reading:**

I made a quick assessment of this paper.

---

> ### Author Response · Authors · 2019-11-09
> **Response to Official Blind Review #3**
>
> Thank you for your comments.
>
> As far as I understand, your belief that our observations might not be true for larger search space is based on the hypothesis that a larger search space makes the hypernet robust and training a child model might not perturb the overall ranking by too much. First of all, this is an educated guess that still requires further experiments or theoretical analysis. Secondly, if larger search space were to be easier, it implies that weight sharing only works on large search space, but we found recently a paper submitted to ICLR (https://openreview.net/forum?id=SJx9ngStPH) which has revealed the instability of weight sharing on large search space (Section 5.2).
>
> We admit that our search space is limited and evaluation on a large search space is an experiment missing in our paper. When developing this small search space, we follow a basic assumption is that smaller search space is easier to search, as intuitively, smaller search space is easier to optimize and very likely an easier problem for weight sharing. If weight sharing doesn't even work on an easy problem, it doesn't make any sense it will work on a harder one.

---

### Decision · Program_Chairs · 2019-12-19

**Decision:**

Reject

**Comment:**

This paper provides a series of empirical evaluations on a small neural architecture search space with 64 architectures. The experiments are interesting, but limited in scope and limited to 64 architectures trained on CIFAR-10. It is unclear whether lessons learned on this search space would transfer to large search spaces. One upside is that code is available, making the work reproducible.

All reviewers read the rebuttal and participated in the private discussion of reviewers and AC, but none of them changed their mind. All gave a weak rejection score.

I agree with this assessment and therefore recommend rejection.